# Analysis of Elastic Properties of Al/PET Isotropic Composite Materials Using Finite Element Method

**DOI:** 10.3390/ma15228007

**Published:** 2022-11-12

**Authors:** Yu-Jae Jeon, Jong-Hwan Yun, Min-Soo Kang

**Affiliations:** 1Department of Medical Rehabilitation Science, Yeo-ju Institute of Technology, Yeoju-si 12652, Korea; 2Mobility Materials-Parts-Equipment Center, Kongju National University, Cheonan-si 31080, Korea; 3Division of Smart Automotive Engineering, Sun Moon University, Asan-si 31460, Korea

**Keywords:** aluminum powder, finite element method, numerical analysis, polyethylene terephthalate, volume fraction

## Abstract

This study uses the finite element method and numerical analysis to develop an eco-friendly composite material with shielding capabilities. A preliminary study was performed to predict the mechanical properties of the composite material. Polyethylene terephthalate and aluminum powder (AP) were selected as the matrix and enhancer, respectively. The particles of AP are spherical, with a diameter of 1 μm. Material properties were investigated as the AP volume fraction (VF) increased from 5–70%. The FEM results show that the physical properties for AP VFs improve by up to 40%, but there is no significant change in the elastic modulus, shear modulus, and Poisson’s ratio at an AP VF of 50–70%. However, the numerical analysis models show that the elastic properties for AP VFs improve by up to 70%. The mechanical properties improved as the VF increased, and the FEM predicted values were reliable for VFs up to 40%. However, it was confirmed that 40% is the limit of AP VF in the FEM. In addition, the FEM and numerical analysis predictions showed that the most similar numerical analysis model was the Halpin–Tsai model. The predictions of the Halpin–Tsai model allowed prediction of the maximum VF above the FEM limit. If the correction coefficients of the FEM and numerical analysis models are derived based on the predictions of this study and future experimental results, reliable predictions can be obtained for the physical properties of composite materials.

## 1. Introduction

High-performance polymers have already been used in several different fields such as automobiles, electrical and electronics, tableware, and construction. Industries are making efforts to develop composite materials such as high-functioning polymers with excellent mechanical properties that cost and weigh less than metals. The use of composite materials is slowly emerging from the realm of advanced materials [1,2,3], allowing them to become increasingly popular in both academic and industrial fields such as automotive, wind energy, aeronautics, and civil applications [4,5]. Recent research on the development of composite materials has focused on achieving multifunctionality and reducing the weight of such materials, and numerous studies have been conducted to improve the mechanical properties of the existing polymer materials [6]. Polyethylene terephthalate (PET), a raw material for high-functioning polymers, is a thermoplastic resin that can be recycled. The reuses of PET waste are essential, preventing environmental pollution, and limiting the exploit plastics derived from petroleum [7]. In addition, PET has excellent mechanical properties and a Young’s or elastic modulus of 3.7 GPa. The main applications of this polymer are the manufacturing of films, fibers, and in the fabrication of bottles for beverages [8]. Future society is demanding characteristics such as miniaturization, high integration, and weight reduction, for mechanical and electronic components. In addition, everyday spaces such as unmanned vehicles are becoming smarter with recent developments in information and communication technology. In daily life, people are never free from exposure to electromagnetic radiation, which may also induce machines to malfunction. An ideal electromagnetic shielding material must offer effective protection from strong, variable electromagnetic fields and have good mechanical properties, along with being light and thin. In addition to possessing good electrical conductivity, the shielding should be adaptable to various environmental conditions, that is, the material should offer high temperature resistance, high pressure resistance, corrosion resistance, dust resistance, and high explosion resistance [9,10,11]. Therefore, many studies have been conducted on the suitability of materials for shielding electromagnetic radiation [12,13,14,15]. With the aim of enhancing the shielding effect, Hung [16,17] used sputtered Sn-Al and Sn-Cu thin films and studied the crystallization mechanism, the impact of film thickness on electromagnetic interference (EMI) properties, and the shielding effect against EMI by coating a composite colloid containing a mixture of Sn-Al powder and polyethylene on glass. The study confirmed that the relationship between coating thickness and shielding was constrained by certain factors. Furthermore, Lee [18] verified the shielding performance through powder coating using mixed carbon black and Zn-Al by thermal spraying. Wang [19] showed that screen-printed carbon nanotube (CNT) sheets can provide effective EMI shielding because they enable better electron transmission compared to that offered by graphite and carbon black sheets. Although the above surface treatment methods can enhance the characteristics of the shielding material and manufacturing processes, the material is still vulnerable to abrasion, peeling, and oxidation. Additional procedures are required to prevent aging of the material, and the maintenance and repair costs are high. There are reports on the verification of shielding materials with a sandwich- or porous structure using Al-based materials [20,21,22]. Composites with discontinuous and continuous conducting fillers, such as metal fibers, metal flakes, carbon particles, and carbon fibers are widely used for EMI shielding [23,24,25,26,27]. However, it is not always easy to determine the suitability of the mechanical properties of such composite materials for use as a conventional shielding structure. Therefore, traditional EMI shielding materials such as permalloy, nickel, and steel are still in use [28]. The possibility of combining two or more materials with different electrical and mechanical properties to obtain an enhanced material with a high grade of homogeneity aligns with the actual trends followed globally, considering the economy and environment [29]. Therefore, studies have been conducted to develop materials with suitable physical properties by using polymers as a matrix with different reinforcement materials such as metals or ceramics. These composite materials may exhibit a continuous or discontinuous mechanism depending on the shape of the reinforcement used, such as fiber, spherical, particle, or flake. In addition, molding and processing according to its use may be difficult, even if the mechanical properties are known. Therefore, it is crucial to be able to predict the physical properties through micro-mechanics. There are several classical methods that exist to accomplish this. The well-known models that have been proposed and used to evaluate the properties of unidirectional composites are the Voigt [30] and Reuss [31] models. The rule of mixture (ROM) equation was derived through these two models based on the volume fraction (VF) of the composite components. Semi-empirical models followed such as the modified ROM (MROM), the Halpin–Tsai model [32], and the Chamis model [33], which were developed to modify the ROM model. The characteristics of the physical properties of the two materials can be predicted by numerical analysis through such models. Alternatively, it is possible to predict the characteristics of the properties of the mixture through the finite element method (FEM) by applying the homogenization technique. As mentioned previously, there are several studies that are aimed at improving the strength of PET materials, realizing lightweight materials, and enhancing shielding effects and mechanical properties through structural changes and usage of alloys based on metals and Al. Furthermore, studies to improve the shielding effects by covering metal or ceramic-based structures with thin films and powder are being conducted. However, the research on the development of composite materials using an eco-friendly material as the matrix and Al as the enhancer, which is lighter than metals and possesses shielding properties, is insufficient. Therefore, this preliminary study performs numerical analysis and finite element analysis to develop a shielding material with high electrical conductivity using reusable PET, acquire lightweight shielding materials using Al, ensure processability that is suitable to the purpose, and enable formability (injection molding) that allows mass production. This study is expected to provide fundamental data for predicting the changes in mechanical properties with AP volume fraction when two materials are mixed through experiments and developing composite materials that are lighter than conventional metal shielding materials.

## 2. Materials and Methods

### 2.1. Workflow Chart

The flow chart for this study is shown in Figure 1. First, PET and Al were selected, and then the volume fraction of Al was selected. Second, finite element analysis and numerical analysis were performed according to the volume fraction of Al. In this case, Ansys mechanical workbench 2021 software was used for finite element analysis, and Reuss and Halpin–Tsai model were used for numerical analysis. Third, the elastic modulus, shear modulus, and Poisson’s ratio calculated from FEM and numerical analysis were analyzed. Fourth, we will plan to continue research on correction factors based on the predicted values calculated from FEM and numerical analysis.

### 2.2. Analysis Model

In this study, the representative volume element (RVE) was modeled for finite element analysis by applying the homogenization method to the PET-based AP, as shown in Figure 2. Here, each colored particle represents an irregularly dispersed AP particle. The generation of RVE plays an important role when determining the effective characteristics of composite materials by applying the FEM using the homogenization method [34]. The RVE representation consists of defining the different supposed homogeneous phases, and in particular their shape, orientation, and distribution. Isotropic composite materials can be manufactured because spherical and particle-like additives are freely dispersed inside the matrix [35,36,37,38,39]. Such isotropic composite materials have no directionality and are thus likely to be widely used in the manufacturing field. Furthermore, the analysis model in this study can predict mechanical properties based on the assumption that the two substances are fully coupled, and the model can reliably interpret particle shape and directionality in the future based on theoretical experiments.

AP was assumed to be a spherical shape with a diameter of 1 μm to reflect the characteristics existing as particles in the matrix material. The size of the RVE model was set at 8 × 8 × 8 μm. The Monte Carlo method [40,41] was used as the code for particle generation to define the position between the particle, particle surface, and collision detection. The Monte Carlo method or experiment is an algorithm that mathematically approximates the value of a function using repeated random sampling. The particles have a constant probability distribution for each position based on a random space in the matrix according to the Monte Carlo method. In addition, homogeneity for each material is deemed to be secure because the position of the particles is calculated using the Monte Carlo method. Table 1 shows the physical properties of PET [42,43] and AP [44,45] to predict the change in physical properties through the FEM and numerical analysis.

### 2.3. Homogenization Method

Experimental analysis for the development of composite materials should consume a lot of time and cost. Therefore, many researchers have been analyzed using finite element analysis [46,47]. In the medical field, there are places where a 2D model with a simple calculation is adopted and analyzed to avoid the 3D shortcomings [48]. In this study, the mechanical properties of composite materials were analyzed by performing 3D finite element analysis through the homogenization method. Homogenization techniques have received increasing attention for predicting mechanical properties of composite materials over the past decades [49,50,51], because homogenization techniques can efficiently quantify the interplay between the microscopic and macroscopic properties [52,53,54]. Ansys mechanical workbench 2021 software was used for the FEM, to which the homogenization technique was applied to calculate the effective material properties of the RVE in which AP, which is the reinforcing agent, is mixed with a PET matrix. The analysis model in which the VF of the reinforcement was set at 5–70% to investigate the material properties as a function of the reinforcement content is shown in Figure 3. Table 2 shows the boundary conditions for the FEM. The boundary conditions of composite materials for predicting physical properties are identical. The element type is SOLID187, and the mesh size is 0.5 μm. SOLID187 shows a quadratic displacement behavior and is suitable for modeling irregular meshes. The matrix and the particle sizes were generated under identical conditions to minimize the sensitivity of physical property prediction according to mesh size. The mesh size was also analyzed under the same conditions to minimize any uncertainty that could stem from the mesh size. The VF for which the most 1 μm AP particles can fit within the size of the RVE model (8 μm × 8 μm × 8 μm) is 52%. Therefore, the VF was set at 60% or more to ensure that the maximum was exceeded.

### 2.4. Numerical Analysis Model

The Voigt model [55,56,57] assumes the uniformity of strain and the Reuss model [58,59] assumes uniformity of stress in composite materials. These two classical bounds are the simplest approaches used to compute the effective elastic properties because they only require a little data at the microstructure level. In the Reuss model, the stiffness of materials is usually measured as the elastic modulus *E*. In the macroscopic elastic range, stiffness is the force required to cause a unit of displacement. Its reciprocal 1/*E* is called the compliance, which is the displacement caused by a unit of force. The Reuss model can be calculated from:(1)$$E_{c}={\left(\overset{n}{\underset{r=0}{\sum }}C_{r}\frac{1}{E_{r}}\right)}^{-1}$$




$E_{c}:\;\mathrm{elastic}\;\mathrm{modulus}\;\mathrm{of}\;\mathrm{composite}\;\mathrm{material}\;\;$



$E_{r}:\;\mathrm{elastic}\;\mathrm{modulus}\;\mathrm{of}\;\mathrm{the}\;\mathrm{material}\;$



$C_{r}:\;\mathrm{volume}\;\mathrm{fraction}\;\mathrm{of}\;\mathrm{the}\;\mathrm{material}$



n: number of synthetic material




The Halpin–Tsai model [60,61] is applied here to predict the compressive elastic and shear moduli of the composite, which are dependent on the particle VF. The Halpin–Tsai approach is simple and easy to use in the design process. The semi-empirical equation can be expressed as Equation (2) below, where ξ is a measure of the particle filler that depends on particle geometry.
(2)$$E_{c}=\frac{E_{0}\cdot \left[E_{1}+\xi \cdot \left(C_{0}\cdot E_{0}+C_{1}\cdot E_{1}\right)\right]}{C_{0}\cdot E_{1}+C_{1}\cdot E_{0}+\xi \cdot E_{r}}$$




$E_{c}:\;\mathrm{elastic}\;\mathrm{modulus}\;\mathrm{of}\;\mathrm{composite}\;\mathrm{material}$



$E_{0}:\;\mathrm{elastic}\;\mathrm{modulus}\;\mathrm{of}\;\mathrm{PET}$



$E_{1}:\;\mathrm{elastic}\;\mathrm{modulus}\;\mathrm{of}\;\mathrm{AP}$



$C_{0}:\;\mathrm{volume}\;\mathrm{fraction}\;\mathrm{of}\;\mathrm{PET}$



$C_{1}:\;\mathrm{volume}\;\mathrm{fraction}\;\mathrm{of}\;\mathrm{AP}$



ξ: measure of particle filler that depends on particle geometry




According to the above numerical method, we calculated the changes in properties of the PET matrix and the AP reinforcing agent as the VF changed from 5–70%.

## 3. Results

In this study, the FEM was performed by selecting PET and AP to develop eco-friendly composite materials with EMI shielding capability. The results are shown in Table 3 and Figure 4. Figure 4a shows the elastic modulus and shear modulus, which are mechanical properties, as functions of AP VF. When the AP VF increases from 5–30%, both the elastic modulus and shear modulus increase approximately linearly. As the VF approaches 40% the rate of increase slows and eventually settles at approximately constant values. At an AP VF of 40%, both the elastic and shear moduli of the composite are more than double the respective moduli of PET. The Poisson’s ratio decreases with increasing AP VF, and at an AP VF of 40% it is approximately 7% less than PET.

Table 4 and Figure 5 show the elastic modulus as a function of the AP VF to compare and analyze the results of the FEM and the numerical analysis. The elastic modulus calculated by the FEM increases with the AP VF until the VF reaches 40%. However, in the numerical analysis results of the Reuss and Halpin-Tsai models, the elastic modulus of the composite material continues to rise as the AP VF increases. At an AP VF of 40% the Reuss and the Halpin–Tsai models predict an elastic modulus that is >1.5 times, and >2.5 times, respectively, that of PET. Note that the Halpin–Tsai model prediction is close to the FEM predictions at a VF of 5–30%. At an AP VF of 70%, the Reuss and Halpin–Tsai model predictions increase to almost three times and over 5 times, respectively, that of PET.

Table 5 and Figure 6 show the shear modulus as a function of the AP VF. Like the elastic modulus, the shear modulus calculated through the FEM increases linearly with the AP VF up to 30% and then gradually levels out. In addition, like the elastic modulus, the shear modulus value continues to increase as the AP VF increases in the numerical analysis results of the Reuss and Halpin–Tsai models. At an AP VF of 40%, the Reuss and the Halpin–Tsai models predict a shear modulus that is >1.5 times and >2.5 times, respectively, that of PET. Note that the Halpin–Tsai model prediction is close to the FEM predictions at a VF of 5–30%. As for the elastic modulus, at an AP VF of 70%, the Reuss and Halpin–Tsai model predictions increase to almost three times and over five times, respectively, that of PET.

Table 6 and Figure 7 show Poisson’s ratio as a function of the AP VF. Poisson’s ratios calculated through the FEM decrease up to an AP VF of 40%; however, the ratio flattens out above 40%. On the other hand, Poisson’s ratios linearly decrease as the AP VFs increase in both the Reuss and Halpin–Tsai models. At an AP VF of 40% the Reuss and the Halpin–Tsai models predict a Poisson’s ratio that is >11% less and >10% less, respectively, that of PET. Note that the predictions of both models are close to the FEM predictions at a VF of 5–30%. At an AP VF of 70%, the Reuss and Halpin–Tsai model predictions decrease to >18% less and >17% less, respectively, that of PET.

In this study, we evaluated the modulus of elasticity, shear modulus, and Poisson’s ratio of the composite materials using the finite element and numerical analyses according to volume fraction with the aim of developing composite materials with good shielding performance and mechanical properties using reusable PET and Al, which are light and possess high electrical conductivity. The FEM results show that there is no significant change in physical properties at VFs of 50–70%. Therefore, it is considered that the effective VF of FEM is limited to 40%. To confirm this the random and theoretical VFs need to be compared. The theoretical VF can be arranged at a VF of up to 52% when AP particles with a diameter of 1 μm are regularly arranged and filled in a cuboid of 8 μm × 8 μm × 8 μm, as shown in Figure 8. The number of particles can be calculated as 512 (8 × 8 × 8). However, the AP particle arrangement is randomly set during the mixing and molding (injection molding) of both PET and AP powders in the FEM of this study. In addition, the arrangement of AP particles is not uniformly arranged in the actual experiment. Therefore, the FEM elastic modulus does not change significantly from that for a VF of 40%, as shown in Figure 4. The same trend is observed for the shear modulus and Poisson’s ratio. Therefore, the limit to which AP particles can be placed without overlapping each other in the RVE model space is 40% of the AP VF.

According to the numerical analysis results in Figure 4, Figure 5 and Figure 6, it is possible to predict until the AP VF is 70% using the Reuss and Halpin–Tsai models. Because the numerical analysis is calculated depending only on the VF without considering environmental factors, it is theoretically possible to calculate for AP VFs of up to 100%. The maximum VF was limited to 52% owing to the sphere-shaped reinforcement used in this study. However, the maximum VF can be calculated differently depending on the shape of the reinforcement. The weight fraction is an easier measure to us in industrial environments while maintaining the isotropy of the composite materials. When the AP VF is 40%, using the densities in Table 1, the weight fraction is 57.5 wt%. In addition, when the AP VF is at the theoretical limit of 52%, the weight fraction is 68.7 wt%. According to a previous study by Adam [62], the shielding performance was excellent when the weight fraction of the reinforcement was 70 wt% or higher. Therefore, to develop an isotropic shielding material using AP and PET in this study, the materials should be developed with an AP VF of 40–50%.

The results of the FEM and numerical analysis need to be compared and analyzed to calculate the predicted values of physical properties outside the range of FEM. Therefore, the results of FEM were compared and analyzed only for the values of the AP VF of 5–40%. The Halpin–Tsai model is a numerical analysis model that was the closest to the FEM predictions of the elastic modulus, shear modulus, and Poisson’s ratio at an AP VF of 5–40%. The predicted value trend line from the FEM limit of 40% of the AP VF to the theoretical maximum AP VF of 52% indicated that the trend calculated using the Halpin-Tsai model was similar to the trend predicted by the FEM. Accurate predictions are limited because experimental values are currently absent. However, the analyzed values that are predicted by the Halpin–Tsai model can be referred to and utilized to confirm and apply the predicted value above the AP VF limit of 40% in the FEM. Correction factors of the FEM and numerical analysis can be obtained using a comparative analysis with experimental values in the future to ensure reliable predictions using theoretical values instead of experimental values.

## 4. Conclusions

In this study, we predict the change of mechanical properties through FEM and numerical analysis as a function of the VF by selecting PET as the matrix and AP as the reinforcing agent. The result is an eco-friendly composite material that has EMI shielding performance and has the lightness, formability, processability, and mechanical properties of composite materials. In addition, we compare and analyze the results.

The FEM results show that the physical properties improve for AP VFs of up to 40%, but there is no significant change in the elastic modulus, shear modulus, and Poisson’s ratio at an AP VF of 50–70%. In addition, the numerical analysis models show that the elastic properties improve for AP VFs of up to 70%. Moreover, when the spherical AP particles are arranged regularly in the RVE model of the FEM, the theoretical maximum AP VF is 52%. However, the maximum effective VF of the FEM is 40% because spherical AP particles are randomly arranged. Therefore, the FEM results show that the elastic properties do not improve beyond the VF of 40%.

When converting an AP VF of 40% using the density, the weight fraction of AP is 57.5 wt%, and 52% converts to 68.7 wt%. Therefore, isotropic shielding materials using AP and PET, which are used in this study, should be developed at AP VFs of 40–50%.

The Halpin–Tsai model is a numerical analysis model with the closest value to the FEM predictions of the elastic modulus, shear modulus, and Poisson’s ratio at an AP VF of 5–40%. Furthermore, the theoretical verification performed in this study is the predicted values of the mechanical material properties calculated under the assumption that AP and PET are completely combined. Therefore, it is necessary in future studies to conduct a comparative analysis through experimental verification to develop the optimal composite materials with shielding performance.

Experimental verifications can analyze the physical properties of AP using various shapes and arrangement structures of AP particles. The methods wherein composite materials are designed according to the weight and volume fractions are required to develop isotropic and formable composite materials. In this study, the limit of theoretical validation for FEM analysis was an AP VF of 40%. Future research should be focused on deriving correction factors for theoretical values based on experimental results, and they should allow calculation of reliable prediction values for the physical properties of composite materials. We aim to establish theoretical correction factors to narrow the reliability gap with experimental values and develop eco-friendly composite materials that can secure mechanical properties and shielding effects according to the shape and arrangement structures of AP particles.

## Figures and Tables

**Figure 1 materials-15-08007-f001:**
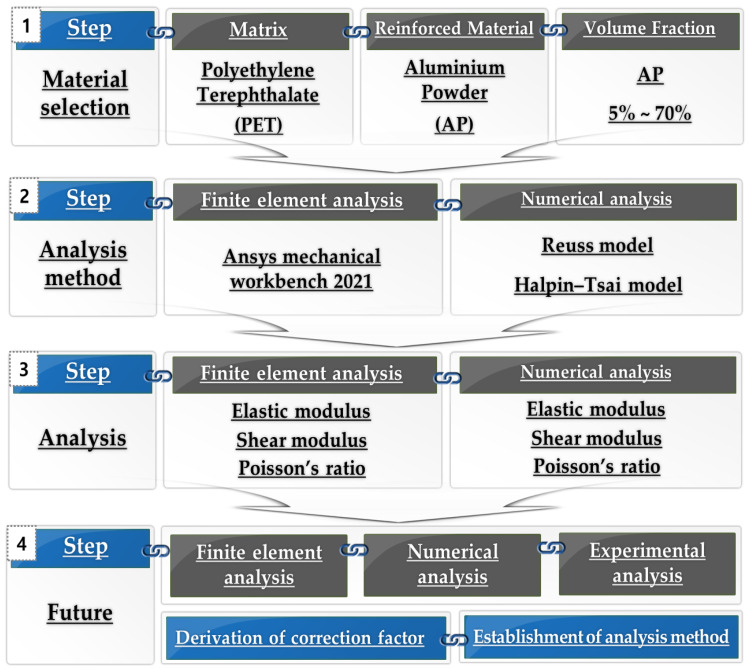
Flow chart for this study.

**Figure 2 materials-15-08007-f002:**
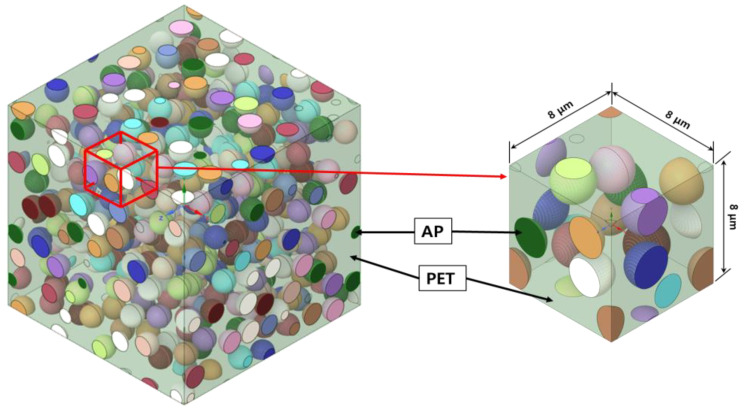
Selection method of representative volume element (RVE) modeling.

**Figure 3 materials-15-08007-f003:**
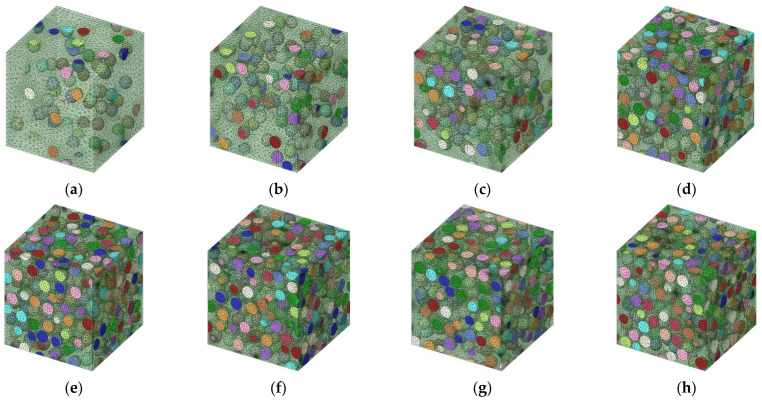
Mesh modeling with AP volume fraction (VF). (**a**) 5%. (**b**) 10%. (**c**) 20%. (**d**) 30%. (**e**) 40%. (**f**) 50%. (**g**) 60%. (**h**) 70%.

**Figure 4 materials-15-08007-f004:**
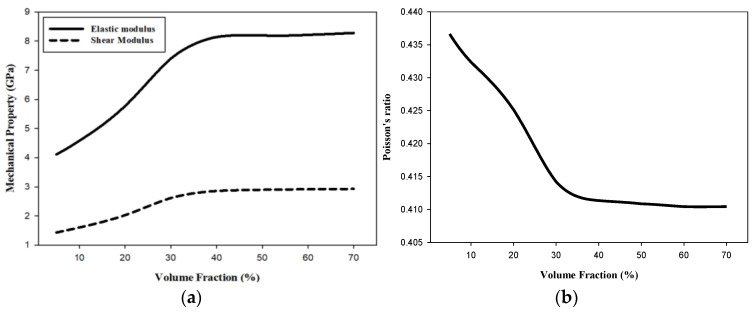
FEM predictions of mechanical properties of composites as a function of the VF. (**a**) Elastic and shear moduli and (**b**) Poisson’s ratio.

**Figure 5 materials-15-08007-f005:**
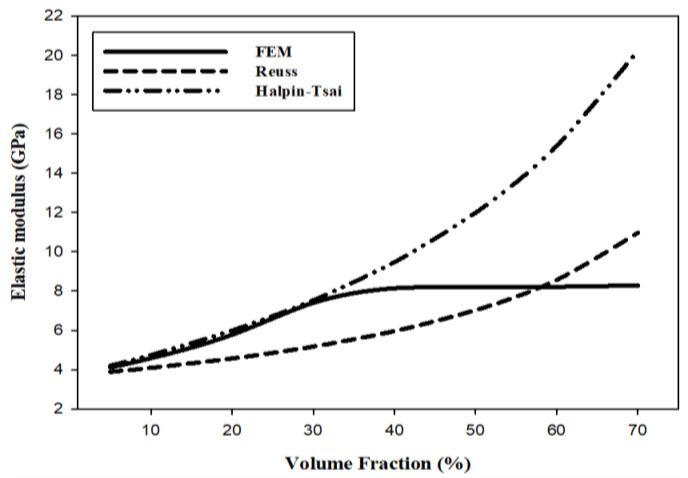
Elastic modulus predictions as a function of AP VF.

**Figure 6 materials-15-08007-f006:**
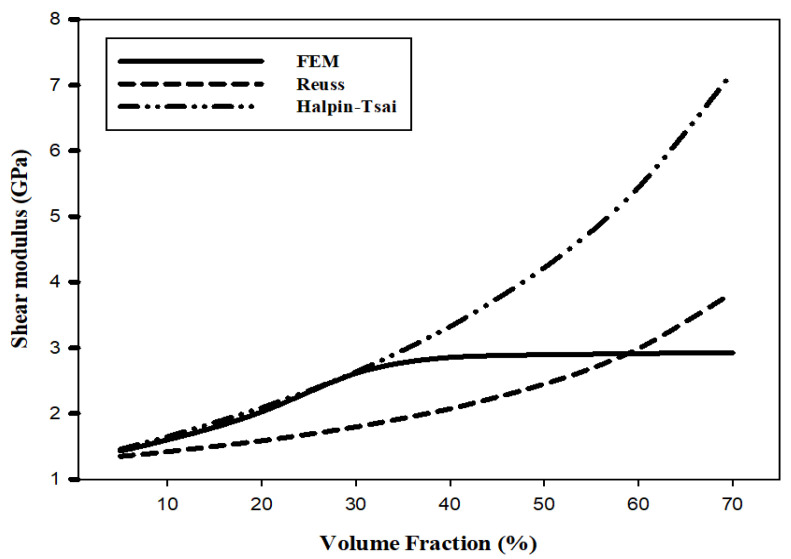
Shear modulus predictions as a function of AP VF.

**Figure 7 materials-15-08007-f007:**
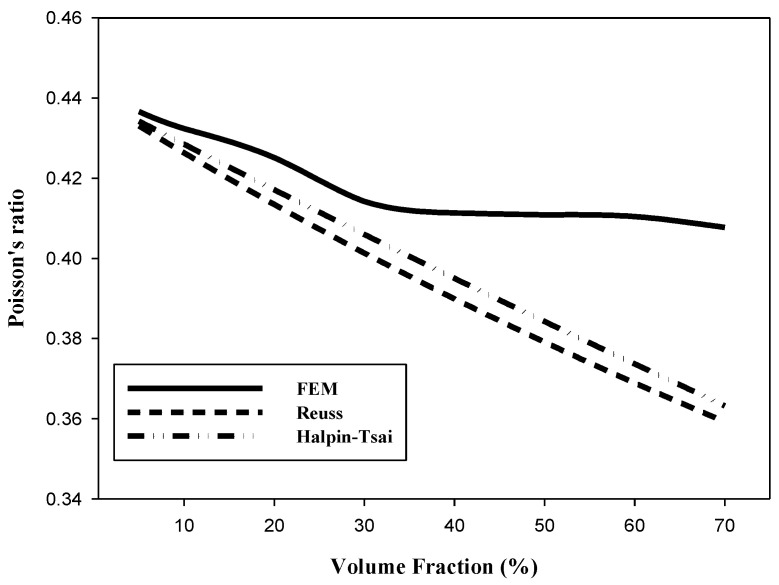
Poisson’s ratio predictions as a function of AP VF.

**Figure 8 materials-15-08007-f008:**
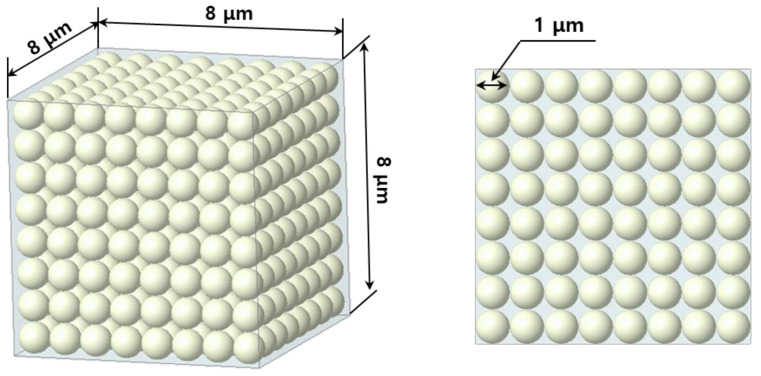
Theoretical arrangement of AP particles in the space of RVE model.

**Table 1 materials-15-08007-t001:** Properties of polyethylene terephthalate (PET) and Al powder (AP).

Properties	PET	AP
Elastic modulus (GPa)	3.700	68.950
Shear modulus (GPa)	1.285	25.863
Poisson’s ratio	0.440	0.330
Density (kg/m^3^)	1330	2700

**Table 2 materials-15-08007-t002:** Boundary conditions per volume fraction (VF).

AP VF (%)	5	10	20	30	40	50	60	70
Node (Ea)	64,228	253,638	383,873	485,350	497,062	497,698	523,377	530,611
Element (Ea)	114,903	177,825	271,362	342,201	350,961	351,389	368,330	372,962
Particles (Ea)	72	145	261	404	447	448	449	456

**Table 3 materials-15-08007-t003:** FEM predictions of elastic modulus, shear modulus, and Poisson’s ratio.

AP VF (%)	5	10	20	30	40	50	60	70
Elastic modulus (GPa)	4.113	4.583	5.773	7.400	8.141	8.197	8.216	8.282
Shear modulus (GPa)	1.430	1.602	2.026	2.613	2.855	2.898	2.916	2.926
Poisson’s ratio	0.437	0.432	0.425	0.414	0.411	0.410	0.410	0.408

**Table 4 materials-15-08007-t004:** FEM and numerical analysis predictions of elastic modulus.

	AP VF (%)	5	10	20	30	40	50	60	70
Elastic modulus (GPa)	FEM	4.113	4.583	5.773	7.400	8.141	8.197	8.216	8.282
	Reuss	3.884	4.087	4.564	5.167	5.954	7.023	8.561	10.961
	Halpin-Tsai	4.195	4.737	5.988	7.527	9.465	11.982	15.382	20.228

**Table 5 materials-15-08007-t005:** FEM and numerical analysis predictions of shear modulus.

	AP VF (%)	5	10	20	30	40	50	60	70
Shear modulus (GPa)	FEM	1.430	1.602	2.026	2.613	2.855	2.927	2.898	2.916
	Reuss	1.349	1.420	1.586	1.797	2.073	2.448	2.989	3.838
	Halpin-Tsai	1.459	1.649	2.090	2.634	3.322	4.219	5.438	7.191

**Table 6 materials-15-08007-t006:** FEM and numerical analysis predictions of Poisson’s ratios.

	AP VF (%)	5	10	20	30	40	50	60	70
Poisson’s ratio	FEM	0.437	0.432	0.425	0.414	0.411	0.410	0.410	0.408
	Reuss	0.433	0.426	0.413	0.401	0.390	0.379	0.369	0.359
	Halpin-Tsai	0.434	0.428	0.417	0.406	0.395	0.384	0.374	0.363

## Data Availability

Not applicable.

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
