# Peer review of "Analysis of Elastic Properties of Al/PET Isotropic Composite Materials Using Finite Element Method"

_materials, 2022, doi:10.3390/ma15228007_

Round 1
Reviewer 1 Report
Based on the assumption of isotropy, authors have studied the elastic properties of Al/PET isotropic composites by finite element method and numerical method. The results are well expressed and discussed reasonably. However, please carefully consider the following comments.
1. Specific research results should be given in the abstract.
2. Lines 8-9: “the aim was to develop eco-friendly composite materials with electromagnetic shielding capability”, This research goal is not supported in the paper. Please provide supplementary and explanation.
3. Please clarify the meaning of each color particle in Figure 1.
4. The source of the material properties in Table 1 should be specified, whether tested, hypothetical, or docum2ented.
5. The meanings of each variable in equations (1) and (2) are not clear and need supplementary explanation.
6. In Figure 3(a), “Gpa” should be changed to “GPa”.
7. The ordinate of Figure 3a, Figure 4 and Figure 5 is wrong.
8. Figure 3, Figure 4 and Figure 5 are duplicated with Table 3, Table 4 and Table 5.
9. It can be seen in Figs. 4 and 5, when the content reaches a certain level, the finite element method differs greatly from the numerical calculation results, and the law also changes. Which results should be followed for this study?
10. Lines 171-174, “In this study, we investigated the elastic modulus, shear modulus, and Poisson's ratio of composite materials as a function of the AP VF, using the FEM and numerical analysis to secure the eco-friendliness, lightness, formability, processability, and mechanical properties of the composite materials.” The supporting evidence for this part is missing.
11. Is it reasonable to assume homogeneity in the study? The homogeneity of materials has a direct effect on their properties. The differential relationship between homogenization and actual materials is missing.
12. The relationship between microstructure and physical properties can be further determined by adding experimental tests and analysis
13. The conclusion needs to be further refined.

Reviewer 2 Report
An estimation of mechanical properties of a composite, where aluminum powder particles used for reinforcement are embedded into polyethylene terephthalate matrix, is presented. The finite element method is employed for the numerical analysis. It is shown how the mechanical material properties of the composite can be improved regulating the volume fraction of powder. Another useful result is that the authors have shown that the Halpin-Tsai model can be employed to determine the effective mechanical properties.
Therefore, I recommend this paper for publication after certain revision. The following points should be adressed:
- Though the results of a theoretical investigation are provided, the source of the mechanical properties of AP and PET should be mentioned in the text.
- The details of the generation of the geometry are not provided for randomly situated particles in the matrix.
- How a certain generation of the stochastic distribution influences of the effective properties? Some details related to structure generation must be provided in the text.
Reviewer 3 Report
1. The title should be changed to use the MDPI format, changing both the uppercase and lowercase characters.
2. The authors need to provide all of the emails after affiliation except for corresponding authors based on MDPI format.
3. At the end of your abstract, please provide a "take-home" message.
4. Sort the keywords according to alphabetical order.
5. It is encouraged not used abbreviations in the keywords section.
6. What is the novel of the present study? It works have been widely studied in the past. Nothing something really new in the present form related to AI/PET composite in computational simulation. The lack of novel seems to make the present study like to replication/modified study. The authors need to detail their novelty in the introduction section. It is a major concern for rejecting this paper.
7. In order to demonstrate the research gaps that the current study aims to address, previous studies linked to it need to be explained in the introduction part, including their work, their novelty, and their limitations.
8. Explain specifically the objective of the present study in the last paragraph of the introduction section.
9. The authors need to explain the advantage of computational simulation via finite element study was conducted in the present paper rather than experimental and analytical investigation such as faster results. Authors must address this crucial aspect in the introduction and/or discussion section. In addition, to support this explanation, the recommended literature should be included as follows: Ammarullah, M. I.; Santoso, G.; Sugiharto, S.; Supriyono, T.; Kurdi, O.; Tauviqirrahman, M.; Winarni, T. I.; Jamari, J. Tresca Stress Study of CoCrMo-on-CoCrMo Bearings Based on Body Mass Index Using 2D Computational Model. Jurnal Tribologi 2022, 33, 31–8. https://jurnaltribologi.mytribos.org/v33/JT-33-31-38.pdf
10. Rather than relying just on the predominate text as it already exists, the authors could incorporate more illustrations as figures in the materials and methods section that illustrate the workflow of the current study.
11. Where is the boundary condition? It is missing.
12. Is the present study performing mesh sensitivity study? It is needs to explained since it would be impacting the results.
13. The authors need to explain meshing strategy in the present finite element model.
14. Derail information regarding number of node, number of element, type of element, threshold, static/dynamic, implicit/explicit, and other should be provided?
15. A comparative assessment with similar previous research is required.
16. At the end of the discussion part, the present study's limitations must be added.
17. Express the conclusion in the form of a paragraph rather than in the current form, that is point by point.
18. Line 91, Ansys version, county, company, needs add related information.
19. Please explain the further research in the conclusion section.
20. The reference needs to be enriched from the literature published five years back, literature published by MDPI is strongly encouraged.
21. The authors sometimes reduced a paragraph to just one or two phrases across the whole article, which made the explanation difficult to follow. To make a more thorough paragraph, the writers should expand upon their explanation. It is advised to include at least three sentences in a paragraph, one of which should serve as the primary idea and the others as supporting details. See line 80-84.
22. Because of grammatical faults and linguistic style, the authors must proofread the document. MDPI English editing service would be a solution.
23. Ensure that the authors followed the MDPI format exactly, edit the current form, and double-check all of the previously noted problems.
Round 2
Reviewer 2 Report
The authors have corrected the paper, so I recommend it for publication.
Author Response
Thank you very much for your review comments.
Reviewer 3 Report
Reviewers greatly appreciate the efforts that have been made by the author to improve the quality of their articles after peer review. I reread the author's manuscript and further reviewed the changes made along with the responses from previous reviewers' comments. Unfortunately, the authors failed to make some of the substantial improvements they should have made making this article not of decent quality with biased, not cutting-edge updates on the research topic outlined. In addition, the author also failed to address the previous reviewer's comments, especially on comments number 6 (nothing something really new), 7 (not in-depth explanation), 9 (suggested reference not included), and 10 (Figure should be provided). For authors feedback in the response, not just simplify mention the rebuttal only mention the line, but also the text in the line to make the reviewer easier to revaluate. With all due respect, the reviewer opposed this article to be published and must be rejected. Thank you very much for the opportunity to read the author's current work.
Author Response
Thank you very much for your depth comments of review. It has been revised according to the review comments. Please review.
